# Gigahertz optoacoustic vibration in Sub-5 nm tip-supported nano-optomechanical metasurface

Renxian Gao[1,4], Yonglin He[1,4], Dumeng Zhang[1], Guoya Sun[1], Jia-Xing He[2], Jian-Feng Li [3], Ming-De Li [2] ✉ & Zhilin Yang [1] ✉

The gigahertz acoustic vibration of nano-optomechanical systems plays an indispensable role in all-optical manipulation of light, quantum control of mechanical modes, on-chip data processing, and optomechanical sensing. However, the high optical, thermal, and mechanical energy losses severely limit the development of nano-optomechanical metasurfaces. Here, we demonstrated a high-quality 5 GHz optoacoustic vibration and ultrafast optomechanical all-optical manipulation in a sub-5 nm tip-supported nano-optomechanical metasurface (TSNOMS). The physical rationale is that the design of the semi-suspended metasurface supported by nanotips of <5 nm enhances the optical energy input into the metasurface and closes the mechanical and thermal output loss channels, result in dramatically improvement of the optomechanical conversion efficiency and oscillation quality of the metasurface. The design strategy of a multichannel-loss-mitigating semi-suspended metasurface can be generalized to performance improvements of on-chip processed nano-optomechanical systems. Applications include all-optical operation of nanomechanical systems, reconfigurable nanophotonic devices, optomechanical sensing, and nonlinear and self-adaptive photonic functionalities.

An optomechanical system is an energy conversion system that uses optically induced thermal, electromagnetic, and optical forces to control motion or mechanical vibrations[1–5]. Rapid progress in micro/nanofabrication technology has resulted in high-quality optomechanical devices with precisely controlled vibrational and optical resonant responses. For example, an optical cavities optomechanical system with an oscillation frequency of kHz-MHz is currently one of the most promising tools for the study of cavity electrodynamics and quantum information[5–7]. With the increasing demand for miniaturization and easy integration of optomechanical systems, nano-optomechanical systems have been rapidly developed and become another vital research branch of optomechanical systems[3,8–13]. With the advantages

of a higher intrinsic oscillation frequency (GHz-THz), easy integration, and lightweight, nano-optomechanical systems have great potential for applications in optical signal processing[10,14], ultrasensitive mechanical sensing[13,15], and coherent phonon quantum control[16–18].

Recently, numerous works on nano-optomechanical systems have been reported; many of these works have been accessible by using ultrafast pulsed laser spectroscopy down to the single-nanoparticle level[8,9,19–22]. This ultrahigh-frequency mechanical oscillation behavior has been shown to be useful for coherent frequency upconversion[8], ultrasensitive high-precision sensing[23], and control of the quantum behavior of phonon modes[9]. With the increasing maturity of meta-material manufacturing technology, metastructure nano-

[1]College of Physical Science and Technology, Xiamen University, Xiamen, China. [2]Key Laboratory for Preparation and Application of Ordered Structural Materials of Guangdong Province, College of Chemistry and Chemical Engineering, Shantou University, Shantou, China. [3]State Key Laboratory of Physical Chemistry of Solid Surfaces, College of Chemistry and Chemical Engineering, Xiamen University, Xiamen, China. [4]These authors contributed equally: Renxian Gao, Yonglin He. ✉e-mail: mdli@stu.edu.cn; zlyang@xmu.edu.cn

optomechanical systems (MNOMSs) have also been rapidly developed. The rich optical modulation ability of metastructures and meta-surfaces opens the possibility of designing nano-optomechanical systems with tailored optomechanical properties and transient all-optical modulation characteristics[12,24,25]. Some breakthroughs have already been made in MNOMSs. For example, Karvounis et al. have provided the first experimental demonstration of giant nano-optomechanical nonlinearity in a plasmonic metamaterial and demonstrated all-optical modulation of transmitted light at 154 MHz[26]. Then, Imade et al. and Dong et al. further scaled the oscillation frequency of MNOMSs to gigahertz[27,28]. Guo et al. demonstrated strong modulation and steering of light via coherent acoustic vibrations in periodic indium-tin-oxide (ITO) nanorod arrays. For the first time, they reported the anisotropic elastic tensor for single-crystal ITO[29]. Recently, Ajia et al. demonstrated a multimodal gigahertz vibrational response corresponding to the mechanical modes of metasurface arrays on a free-standing dielectric membrane[11]. Despite these exciting discoveries, MNOMSs still face some fundamental challenges. First, the acoustic vibration of MNOMSs is limited by multiple energy loss channels, resulting in low oscillation quality. Second, the nonradiative losses of a nano-optomechanical metasurface under the pumping of a high-energy pulsed laser are extremely susceptible to structural damage and transient optical performance degradation[30].

In this work, we demonstrate high-quality-factor gigahertz optoacoustic vibration and ultrafast nano-optomechanical manipulation in a sub-5 nm tip-supported nano-optomechanical metasurface (TSNOMS). The main idea for overcoming these challenges is to close the main loss channels for the optoacoustic vibration of the nano-optomechanical metasurface and find mild optical forces excitation modes. The semi-suspended state design of TSNOMS opens the optical input energy channel and closes the mechanical and thermal output loss channels, thus significantly improving the optomechanical conversion efficiency and oscillation quality of the optomechanical metasurface. In addition, the construction of a nonlocalized interband-transition-induced optical forces excitation mode greatly improves the optomechanical conversion efficiency while avoiding structural

damage and transient optical performance degradation caused by nonradiative losses of the metasurface. The multichannel-loss-mitigating semi-suspended metasurface design strategy can be generalized to performance improvements of most on-chip processed nano-optomechanical systems. And the strategy will drive the development of steady-state metamaterials towards reconfigurability, opening exciting prospects for all-optical operation of nanomechanical systems, reconfigurable nanophotonic devices, optomechanical sensing, and novel nonlinear and self-adaptive photonic functionalities.

## Results and discussion

The sub-5 nm TSNOMS was designed as a potential high-quality optomechanical resonator and ultrafast optomechanical all-optical modulator. As shown in Fig. 1a, b, the structure consists of a sharp silicon nanotip array and a semisuspended ultrathin gold nanohole metasurface. An inductively coupled plasma (ICP) etching process was used to fabricate the sub-5 nm TSNOMS (see the detailed process flow in Supplementary Fig. 1). Scanning electron microscopy (SEM) images of the TSNOMS are shown in Fig. 1c. Obviously, the experimentally prepared structural morphology is highly consistent with the structural model we designed. The array studied has a periodicity of 500 nm; the nanoholes have an average diameter of 300 nm and a thickness of 20 nm. The metasurface period, diameter, and thickness can be flexibly and precisely regulated over a large area by the poly-styrene sphere nanoarray template etching process. ICP etching allows for the preparation of nanodevices with large areas that are extremely uniform and precise. Thus, the size dispersion of TSNOMS is related only to the homogeneity of gold nanohole array templates. As shown in Fig. 1d, the structure exhibits large areas that are very uniform and aligned, with a relative standard deviation of less than 1% in the hole diameter. (See Supplementary Fig. 2 for more SEM images.) As shown in Supplementary Fig. 2, the silicon nanotip arrays also exhibit great homogeneity over a large area due to the extremely homogeneous nanohole array templates. The excellent uniformity of the size and shape of the TSNOMS minimizes inhomogeneous broadening effects.

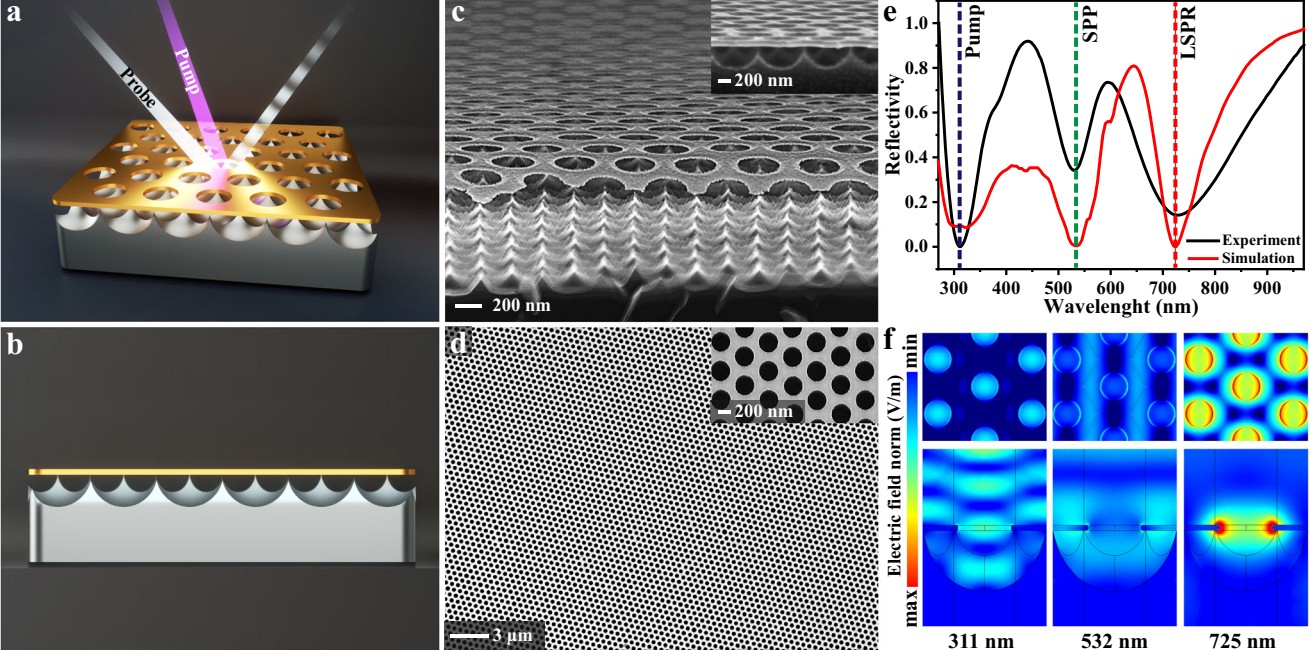

**Fig. 1 | Structural and steady-state optical measurements and simulations.**
**a** Concept image showing the vibrational response of the probe signal through optical excitation of the TSNOMS. **b** Concept image of the TSNOMS. **c, d** SEM image of the TSNOMS. **e** Measured reflection spectra of the TSNOMS (black lines) and reflection spectra simulated by the finite element method (red line). **f** The simulated electric field intensity distribution of the TSNOMS.

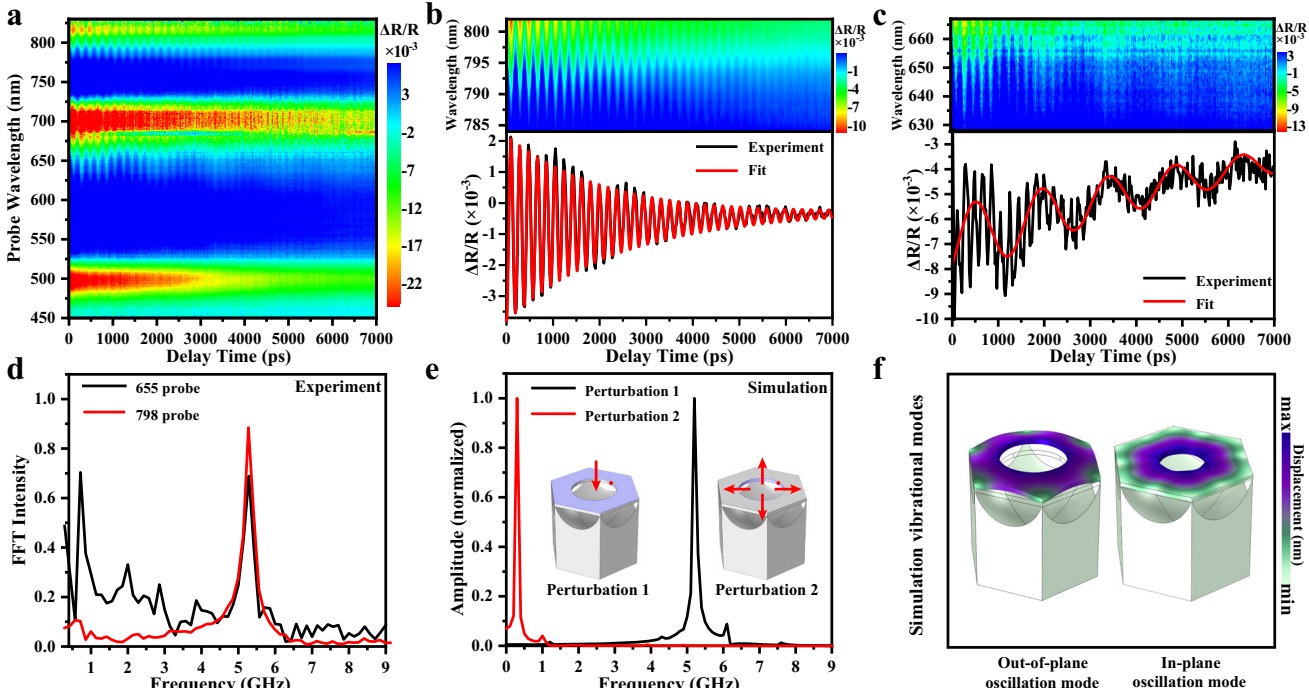

**Fig. 2 | Transient optical measurements and simulations. a** Transient reflection spectral map of the TSNOMS for the spectral range between 450 and 830 nm with a delay time up to 7000 ps. **b** Transient reflection spectral map for the spectral range between 785 and 830 nm and ΔR/R kinetics at 798 nm. **c** Transient reflection spectral map for the spectral range between 630 and 668 nm and ΔR/R kinetics at 629 nm. **d** Fast Fourier transform (FFT) spectra of the ΔR/R kinetics at 629 and 798 nm. **e** Simulated frequency spectra of displacements sampled at a point (indicated by a red dot) on the edge of a nanohole. **f** Simulated mechanical eigenmodes at selected frequencies.

Notably, the tip diameter of the silicon tip is much smaller than 5 nm. This design ensures structural stability while minimizing the contact area between the substrate and the metasurface, which provides the basis for further investigation of the transient optomechanical properties of the TSNOMS.

To correctly assign the optical resonance mode and select the appropriate transient pump wavelength to excite optoacoustic vibration of the nano-optomechanical system, the optical reflectance spectrum of the TSNOMS was explored experimentally and simulatively. The results for the experiment (solid black line) and finite element method electromagnetic simulation (COMSOL Multiphysics, solid red line) are shown in Fig. 1e. Details of the electromagnetic simulation are given in methods and Supplementary Fig. 3. As shown in Fig. 1e, the measured profile is in qualitative agreement with the numerical simulation results. Three dips appear at 311, 532, and 725 nm in the spectrum. To further clarify the intrinsic mechanism of these modes, near-field maps of the electric field intensity are presented in Fig. 1f. The resonance absorption at 725 nm results from the dipolar localized surface plasmon resonance (LSPR) of the nanoholes. The resonance absorption at 532 nm corresponds to the propagating surface plasmons (PSPs) of the metasurface. Near-field maps at 311 nm do not show strong localized electric fields. The absorption at 311 nm is attributed to the interband transition absorption induced by ultraviolet interference[31]. For a detailed analysis of the interband transition absorption, see Supplementary Fig. 4. Based on the above mode analysis, 311 nm was chosen as the pump laser wavelength. The reason is that when the 311 nm pulsed laser interacts with the gold metasurface, a large number of electrons at lower energy levels will be excited to higher energy levels. These non-thermal-equilibrium electrons at high energy levels transfer energy to the lattice, resulting in rapid thermal expansion of the structure[22,32]. We predict that this expansion, in turn, excites coherent acoustic vibration modes that result in periodic geometrical deformation of the nanostructures.

The time-resolved ultrafast spectroscopy technique was used to investigate the GHz optoacoustic vibration modes and the ultrafast nano-optomechanical manipulation of the TSNOMS. A 311 nm pulsed laser (pulse duration of 84 fs) was used as the pump source. The pump beam was collimated with a Gaussian diameter of 1 mm (see methods and Supplementary Fig. 5 for details of the experimental setup). Figure 2a presented the calculated ΔR/R spectral map under a pump fluence of 193.8 μJ/cm². The ΔR/R spectral map contains two major transient characteristics. As for the first transient characteristic, two sets of photoinduced absorption and bleaching oscillate, corresponding to the two static dips at 532 nm (PSPs mode) and 725 nm (LSPR mode) in Fig. 1e. The second transient characteristic has been an oscillation for the ΔR/R signal that is overlaid upon the initial characteristic. Cyclical oscillation for ΔR/R signal has been ascribed to activation of coherent optoacoustic vibrations of the metasurface, which adds strain as well as geometrical distortion, hence altering its optical transmission (see Supplementary Fig. 6 for details of the time domain physical process). To obtain more explicit transient oscillation information, the two sets of transient signals corresponding to the PSPs mode and LSPR mode are highlighted in Fig. 2b, c. One high-frequency oscillation and one relatively low-frequency oscillation can be seen in the transient signals of the LSPR and PSPs modes, respectively. Notably, the distribution of relatively low-frequency oscillation is mainly in the red sidebands of the PSPs mode (approximately 625–630 nm), while the high-frequency oscillations are in the red and blue sidebands of the LSPR mode at approximately 625–630 nm and 775–800 nm, respectively. The reasons for the period modulation of the sidebands of transient spectra are discussed in Supplementary Fig. 7 and Supplementary Fig. 8. The maximal values of ΔR/R are 0.22% at 629 nm and 0.55% at 798 nm for the relatively low-frequency and high-frequency oscillations, respectively, as shown in Fig. 2b, c. Notably, the portion of modulated light is related to the power density of the pumped laser, and thus, we have provided power-dependent

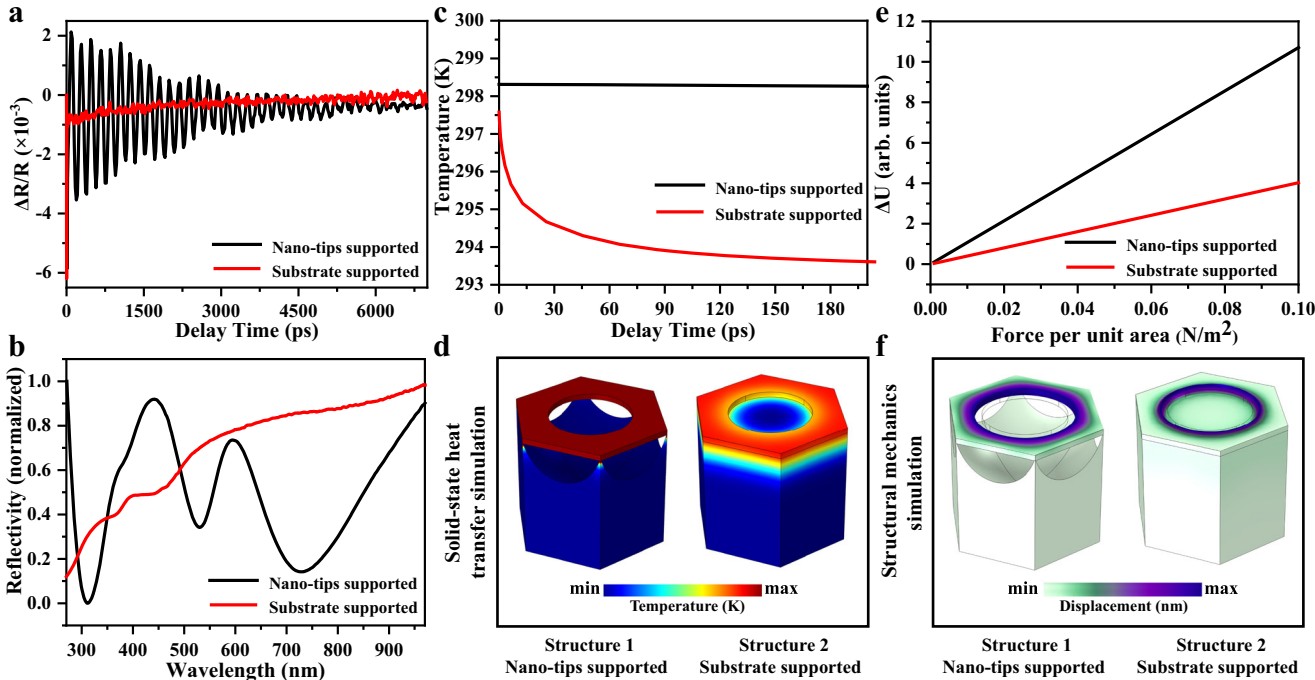

**Fig. 3 | Transient and steady-state optical measurements and thermal, mechanical simulations. a** Transient reflection spectra of the optomechanical metasurface supported on a monocrystalline silicon substrate and the silicon nanotip arrays. **b** Optical reflectance spectra of the optomechanical metasurface supported on a monocrystalline silicon substrate and the silicon nanotip arrays. **c, d** Simulated thermal distribution in the time domain of the two supported forms of the optomechanical metasurface. **e, f** The simulated geometric strain of the metasurface under the same prestress.

transient reflection spectra in the Supplementary Fig. 9. For the present research, experiment traces had been matched by the following function.

$$\Delta I(t) = \sum_{k=(el,ph)} A_k \exp\left(-\frac{t}{\tau_k}\right) + \sum_{n=(1,2,\ldots)} A_n \cos\left(\frac{2\pi t}{T_n} - \phi_n\right)\left(-\frac{t}{\tau_n}\right) \tag{1}$$

Where $\tau_k$ is a period of decay in correlation to the thermal coupling of an environment, $\tau_n$ is a period of damping for optoacoustic vibrations, $T_n$ is the vibration period, $A_k$ and $A_n$ are the amplitudes, while $\phi_n$ is the phase factor. The initial term correlates to a background signal arising from electron-phonon and phonon-phonon interactions. The second term accounts for various vibrations, with $n = 1$, and 2, representing the mode number. In Fig. 2b, c, the high-frequency mode and relatively low-frequency mode were fitted to the equation with periods $T_h = 190$ ps and $T_l = 1360$ ps and damping times $\tau_h = 2240$ ps and $\tau_l = 4500$ ps. Such outcome yields the factor of quality to the high-frequency mode of $Q = \pi\tau_h/T_h = 37$ This $Q$ value is the greatest recorded in terms of metastructure and metasurface optomechanical resonators to date. The information of fast Fourier transform (FFT) has been illustrated in Fig. 2d. The frequencies $f_h = 5.27$ GHz and $f_l = 0.71$ GHz are consistent with the time domain outcomes.

To better understand the physical origin of the transient oscillation signal, structural mechanical mode analysis of the TSNOMS was carried out using the finite element method. Harmonic disturbance boundary conditions were applied in the normal directions of the nanohole film upper surface and the nanohole's inner surface (see the purple marked area of the schematic diagram of the structure in Fig. 2e). The frequency of the harmonic disturbance was calculated by frequency scanning (for details, see methods). Figure 2e shows the simulated frequency spectra of displacements sampled at a point (indicated by a red dot) on the edge of a nanohole. The two prominent peaks at 0.3 and 5.19 GHz are qualitatively consistent with the

experimental observations shown in Fig. 2d. Compared with the simulation values, the experimental FWHM result is relatively wider, which is presumably due to the combined effects of inhomogeneous broadening, overlapping resonances, dephasing of different resonators, and intrinsic damping in the experiment. After confirming the consistency of the experimental and theoretical simulation results, an eigenmode analysis was performed for the TSNOMS with the finite element method. The two most dominant vibration modes were selected based on the fitting results of Fig. 2e and are plotted in Fig. 2f. The purple and green sections represent the maximum and minimum deformations, respectively. According to the numerical calculations in Fig. 2e, f, the transient oscillation signal with the value of 5.19 GHz is caused by modulation of the spectral dispersion by the periodic in-plane deformation of the nanohole. The transient oscillation signal with the value of 0.3 GHz is caused by out-of-plane periodic oscillations of the structure. In addition, the reflectance spectra of the structures after oscillatory deformation were calculated by the finite element method to further confirm the accuracy of the mechanical mode attribution. (For detailed simulation results, see Supplementary Fig. 7 and Supplementary Fig. 8).

Figure 3a shows the transient reflection spectra of the optomechanical metasurface supported on a silicon substrate or the silicon nanotip array under excitation by a 311 nm pump. The support of the metasurface by the nanotip array exponentially improves the metasurface transient signal oscillation modulation ability compared to the silicon substrate support. To understand the reasons for the significant improvement in transient signal modulation, detailed numerical calculations and experimental analyses were performed from the perspectives of the optical, thermal, and mechanical energy channels. For the optical energy channel, the optical reflectance spectra of the metasurface with the two types of support were experimentally measured (Fig. 3b). The TSNOMS exhibits clear optical resonance absorption dips compared to the silicon-substrate-supported metasurface. This allows the TSNOMS to couple the energy of the pulsed

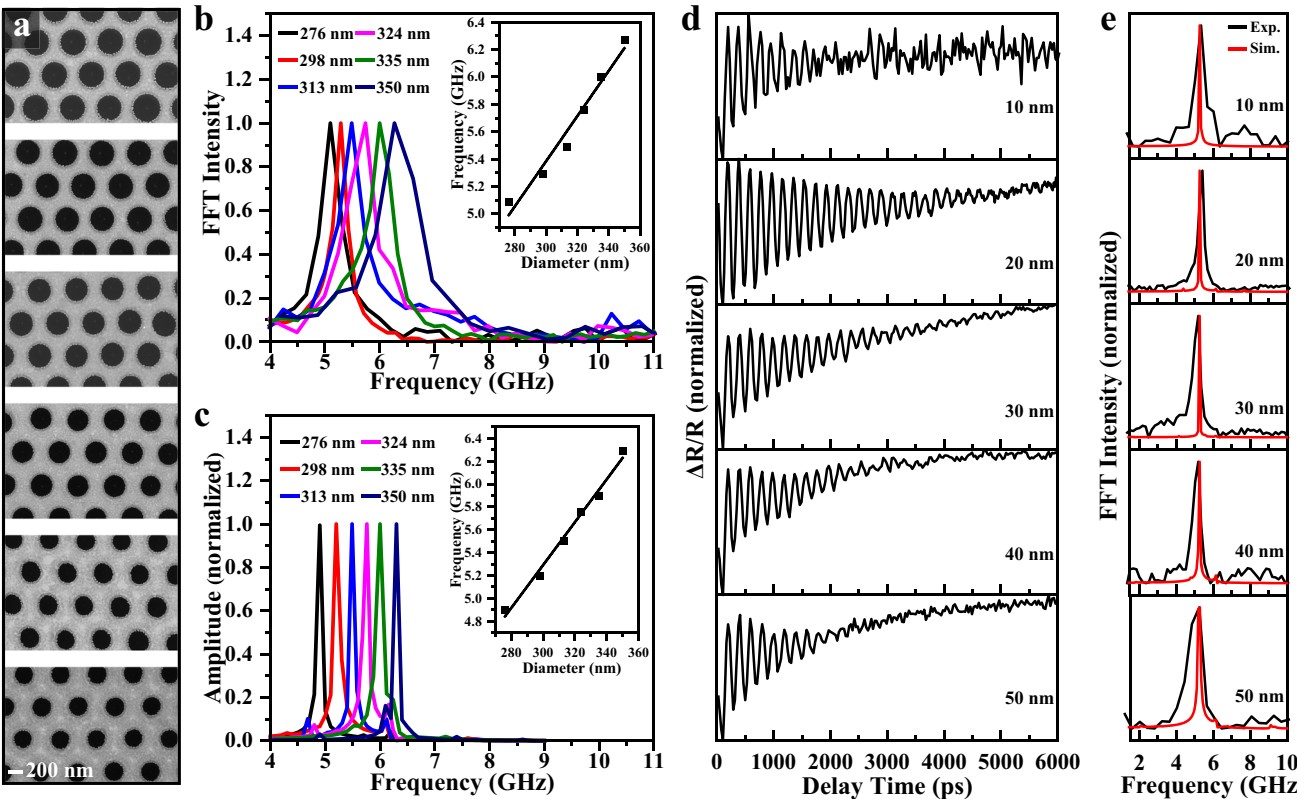

**Fig. 4 | Experiments and simulations of structural parameter modulation.**
**a** SEM images of the TSNOMS with different nanohole diameters. **b** FFT spectra of the measured reflection optomechanical responses with varying diameters of nanohole. Inset: Oscillation frequency as a function of nanohole diameter. No error bars are added since the oscillation frequency tested is almost constant.

**c** Simulated frequency spectra of the TSNOMS with different nanohole diameters. **d** Transient reflection spectra of the optomechanical metasurface with varying thickness of metasurface. **e** FFT spectra of the measured reflection optomechanical responses with different thicknesses. The black lines are experimental data, and the red lines are simulation results.

laser more easily into the optomechanical system to excite optoacoustic vibration. In addition, the optical resonance absorption enables more straightforward observation of the periodic modulation of transient signals by the metasurface. For the thermal energy channel, the thermal distribution in the time domain of the two supported forms of the metasurface was calculated by coupling the solid-state heat transfer model with the two-temperature model (2TM)[33-35] and the wave optics module of COMSOL (for details, see methods). As shown in Fig. 3c, the temperature of structure 2 rapidly decays after 10 ps, whereas structure 1 remains at a relatively high temperature at 200 ps (The electronic and lattice temperatures of the structures at the first 10 ps were shown in Supplementary Fig. 10a). To clarify the reason for such a large difference in the temperature of the two structures, the spatial thermal distribution in the structures at 200 ps was calculated. As shown in Fig. 3d, there is a large amount of heat diffusion into the substrate in structure 2 because of the direct contact between the gold film and the substrate. In structure 1, the ultrasmall contact area between the substrate and the metal nanohole film array dramatically reduces the heat loss of the metasurface. The thermal strain on the tip-supported metasurface can be expected to be more remarkable than that on the silicon substrate metasurface for the same pump conditions. For the mechanical energy channel, the geometric strain of the metasurface under the same prestress was simulated by the finite element method for both support methods (see Fig. 3e, f). The deformation variables of structure 1 are much larger than those of structure 2 under the same prestress. Thus, we can determine that the support of the metasurface by the sub-5 nm tip can close the energy channel for phonon leakage to the substrate to the greatest extent. Therefore, the design strategy of a sub-5 nm tip-supported

metasurface can reduce the energy loss from the substrate to close to the theoretical limit. (See Supplementary Fig. 10 for a detailed discussion of energy losses).

Frequency tunability and frequency stability of optomechanical metasurfaces are essential for optical signal processing, ultrasensitive optomechanical sensing, and photonic functionality metasurfaces. Figure 4a, b shows SEM images of the TSNOMS with different hole diameters (276, 298, 313, 324, 335, and 350 nm) and the FFT spectra. The resonance frequency of the metasurface increases from 4.9 to 6.3 GHz as the hole diameter increases from 276 to 350 nm. There is an excellent linear relationship between the hole diameter and resonance frequency. No error bars are added since the oscillation frequency tested is almost constant. The attribution of resonance modes in Fig. 2f already indicates that the high-frequency transient oscillation signals of the metasurface originate from the in-plane ring breathing oscillations of the nanoholes. The clear oscillation frequency dependence on the hole diameter allows us to easily precisely tune the oscillation frequency through the reactive ion etching (RIE) process. The numerical theoretical calculations in Fig. 4c generally agree with the experimental results. Notably, the transient dynamics signal of mechanical oscillations is completely undetectable on the substrate-supported metasurface due to the enormous losses from the substrate in Fig. 3a. In this sense, frequency tunability is a unique property of TSNOMS arrays. In addition, the thickness dependence of the vibration response was investigated, as shown in Fig. 4d, e. The oscillation frequency of the sample hardly varies with the sample thickness, and the finite element calculations agree with the experimental results. This is because the intrinsic frequencies of the inner-ring breathing oscillation modes are only related to the diameter of the nanohole and not to the

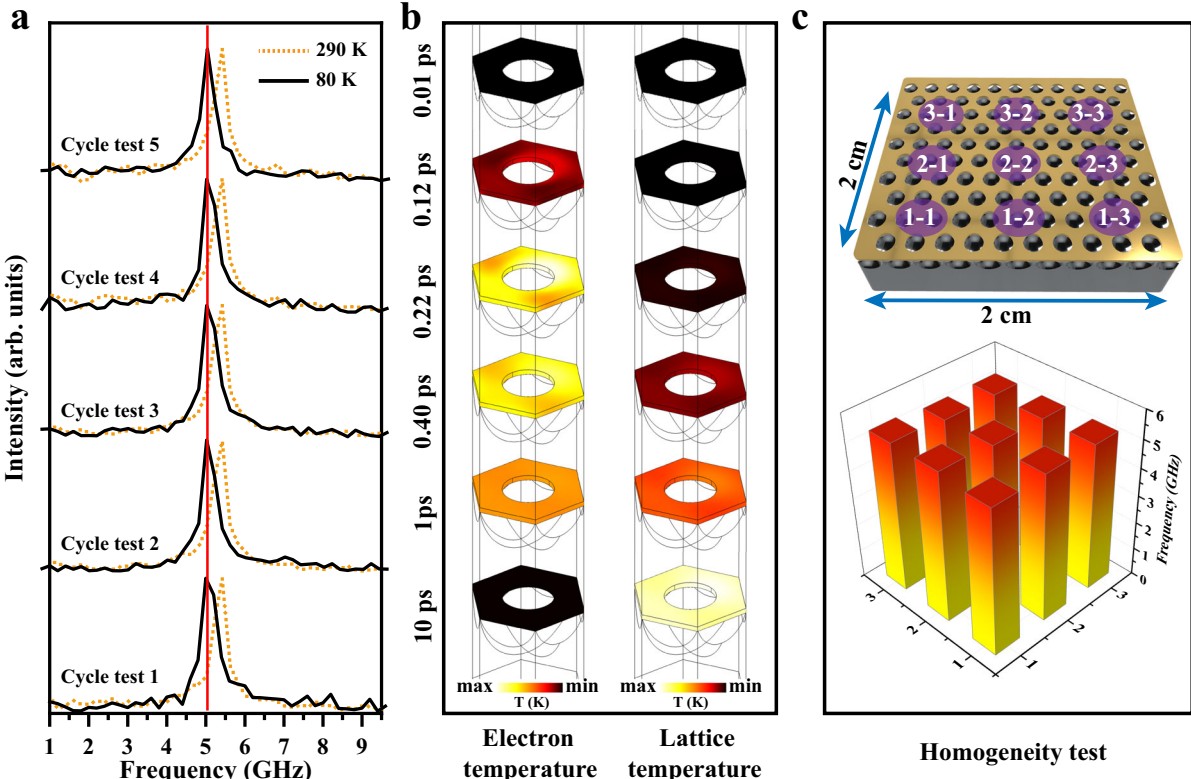

**Fig. 5 | Stability measurements. a** Oscillation FFT spectrum of the TSNOMS under 5 temperature change cycles. **b** Simulated thermal distribution of electron and lattice temperatures on the TSNOMS. **c** Illustration of the aerial pump-probe scan of the TSNOMS with labeled scan regions and histogram of resonance frequencies corresponding to the labeled scan regions.

thickness. This characteristic provides us with a frequency stability guarantee even when specific processing thickness errors occur during the processing of the sample.

Finally, we tested the stability of the metasurfaces in terms of several areas, including their low-temperature tolerance, in-situ cycling stability, and homogeneity. For the low-temperature tolerance, the vibration performance of samples under 5 temperature change cycles was tested in-situ using a transient vacuum cryogenic insert (testing temperatures of 80 and 300 K). Figure 5a shows the oscillation fast Fourier transform (FFT) frequency spectrum obtained by in-situ tests under five variable temperature cycles. A blueshift of the resonance frequency is found to accompany the decrease of temperature due to the stiffening of the lattice[36]. Notably, the samples oscillate at a highly consistent frequency in the 80 K environment over the five cycles of the test. This low-temperature tolerance is valuable for the sensor or transient optical modulation in harsh environments. For the in-situ cycling stability, normally, metal metasurfaces are highly susceptible to damage due to the nonradiative losses of metals under the action of high energy pulsed lasers. However, as shown in Fig. 5a, the resonance frequency and half-peak width of the TSNOMS remained almost unchanged after 10 in-situ tests, which far exceeded our expectations. Therefore, the thermal distribution of the TSNOMS under a 311 nm pump was calculated by the wave optics module of COMSOL with the 2TM, as shown in Fig. 5b. The transient heating of the metasurface by 311 nm pump light results from nonlocal heating via interband transition excitation; thus, the electron and lattice temperature distributions on the metasurface are not highly localized. The construction of a nonlocalized interband-transition-induced optomechanical conversion mode avoid structural damage and transient optical performance degradation caused by nonradiative losses of the

metasurface. In terms of homogeneity, we tested the uniformity of the signals of the samples. This is explored in Fig. 5c, presenting a set of pump-probe measurements over a 2 × 2 cm area. The corresponding time-domain plots and resonance frequency histogram for each indicated point are shown in Fig. 5c and Supplementary Fig. 11. The oscillation frequencies of the samples are relatively uniform. The relative standard deviation of the oscillation frequency is 1.1%. Since the oscillation frequency tested at the same point is almost constant, no error bars are added.

In conclusion, we demonstrated a high-quality 5 GHz optoacoustic vibration and ultrafast optomechanical all-optical manipulation under UV band pumping in a sub-5 nm TSNOMS. The design of the semi-suspended metasurface supported by nanotips of less than 5 nm cleverly enhances the optical energy input into the metasurface and closes the mechanical and thermal output loss channels of the metasurface, thus significantly improving the optomechanical conversion efficiency and oscillation quality of the metasurface. The quality factor has been improved by orders of magnitude compared to normal silicon-substrate nanohole arrays. Furthermore, the TSNOMS allows precise manipulation of arbitrary structural parameters and accurate adjustment of the oscillation frequency of the sample using a low cost, large area RIE template etching method. In addition, the structure can maintain in-situ cycling stability over significant temperature variations from 80 to 270 K, which enables stabilized high-performance sensing and transient all-optical modulation in complex temperature environments. The design strategy of a sub-5nm tip-supported metasurface can reduce the energy loss from the substrate to close to the theoretical limit. This strategy will drive the development of steady-state metamaterials towards reconfigurability and facilitate further nanosizing and integration of optomechanical systems.

## Methods

### Time-resolved experiments

The ultrafast dynamics for TSNOMS were monitored by the femtosecond optical pump-probe technique. The 84-fs amplified titanium sapphire laser (Coherent, Astrella-Tunable-F-1k) was deployed function for the pump source. The optical parametric amplifier (OPA-800CF, Spectra-Physics) in a combination with a harmonic module was deployed to change the pump source wavelength to 311 nm. A variable ND filter was used to manage the output of the pump collimated pump beam with a Gaussian diameter of 1 mm was used to excite the vibration modes of the samples laser probe pulse was produced with ~4% of amplified 800 nm laser pulses to generate a white-light continuum (320–850 nm) in a sapphire crystal, and then, this probe beam was split into two parts. The sample would receive one beam, while the reference spectrometer would receive another to ascertain the intensity. The pump-induced relative changes in the probe beam reflection, ΔR/R, were measured using synchronous detection as a function of the time interval separating the pump and probe pulses, controlled by a mechanical delay line. A schematic diagram of the femtosecond transient reflection spectroscopy is shown in Supplementary Fig. 5.

### Optical simulations

We adopted the finite element method commercial software (COMSOL Multiphysics) to calculate the far-field reflectance spectra and local-field distributions of the TSNOMS. The dimensions of the modeled system were chosen to match the previously designed structure. The optical constants of gold and silicon were taken from the material library. A port condition was configured to generate plane waves traveling in the z-direction with electric-field polarization along the x-axis (with an incident power of 1 W). On the z-directional surfaces of the simulation domain, periodic boundary conditions were applied in the x-y plane. A total of approximately 120,000 tetrahedral mesh elements were used for discretization. Details of the simulations are given in Supplementary Fig. 3.

### Mechanics simulation

The COMSOL structural mechanics module was used to simulate the mechanical modes of the TSNOMS. Eigenfrequency and frequency domain solvers were employed to extract specific eigenfrequencies as well as the full frequency spectrum of the mechanical modes. Measurements for the modeled system were selected to match optical simulations. Periodic boundary conditions were applied in the x-y plane. The mesh size was chosen after convergence analysis for convergence. Bulk properties for gold were selected as young's modulus corresponds to 79 GPa, with density at 19.3 g/cm³ while passion ratio at 0.44. Furthermore, bulk properties of silicon were selected while young's modulus was at 170 GPa, with density at 2329 kg/cm³ while passion ratio at 0.28 (from the material library of the software).

### Temperature dynamics simulations

The solid-state heat transfer module and general form partial differential equation module of COMSOL and the two-temperature model (2TM) were used to investigate the temperature dynamics processes arising from a femtosecond laser pulse interacting with a metasurface[33–35]. We divided the temperature dynamics simulation into two steps. Step 1 involved calculating the temperature dynamics processes from 0 to 10 ps after pulsed laser irradiation of the metasurface. In this part, the electromagnetic loss distribution Q (x, y, z) in the gold nanostructure was calculated by the wave optics module in the frequency domain. This loss was then multiplied by the time profile of the laser pulse intensity to obtain the heat source term in the two-temperature model equation. The intensity profile of the femtosecond laser pulse was assumed as Gaussian and the peak power of the pulse reaches the metasurface at $t_0 = 0.16$ ps. Then, we simulated the

evolution of electron and lattice temperatures inside the gold by solving 2TM with general form partial differential equation module of COMSOL. As the lattice thermal conductivity of gold is much smaller than the electronic thermal conductivity, the temperature exchange between the metasurface and the substrate was neglected within 0–10 ps. Therefore, the 2TM calculations were performed only in metal nanostructures. The 2TM consists of two equations describing the electron temperature $T_e$ and the lattice temperature $T_l$.

$$C_e(T_e)\frac{dT_e}{dt} = \nabla \cdot (\kappa_e \nabla T_e) - G(T_e)(T_e - T_l) + S(r,t) \quad (2)$$

$$C_l \frac{dT_l}{dt} = \kappa_l \nabla^2 T_l + G(T_e)(T_e - T_l) \quad (3)$$

with $C_e$, $\kappa_e$, $C_l$ and $\kappa_l$ for electron and lattice heat capacity and thermal conductivity, $G$ for electron-phonon coupling constant. $S(\mathbf{r},t)$ for the heat source. $C_e = \gamma T_e$, $\gamma = 71 J/(m^3 \cdot K^2)$, $\kappa_e = \kappa_0 T_e/T_l$, $\kappa_0 = 317 W/(m \cdot K)$, $C_l = 2.492 \cdot 10^6 J/(m^3 \cdot K)$, $\kappa_l = 2.6 W/(m \cdot K)$ and $G = 2.2 \cdot 10^{16} W/(m^3 \cdot K)^{[37–39]}$. Step 2 involved calculating the dynamical processes of the metasurface after 10 ps. Since the electron and lattice temperatures converge after 10 ps, the solid-state heat transfer module was chosen to calculate the temperature exchange dynamics between the metasurface and the substrate. In this part, the initial temperature of the gold in the metasurface was set to the spatial average of the lattice temperature at the end of step 1. The initial temperatures of the substrate and air were set to $T = 293$ K.

## Data availability

The source data that support the findings of this paper are provided with this paper. Source data are provided with this paper.

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

## Acknowledgements

This work was supported by the National Natural Science Foundation of China (12174324, Y.Z. and 92161118, Y.Z.) and the National Key Research and Development Program of China (2021YFA1201502, Y.Z. and 2017YFA0204902, Y.Z.).

## Author contributions

G.R. and H.Y. conceived the idea. G.R. manufactured the samples and coordinated overall research effort. G.R., H.Y., and H.J. performed the pump-probe experiment. G.R. and Z.D. performed the numerical simu-lations. H.Y., Y.Z., L.M., L.J., and S.G. supervised the experimental work. G R., Y.Z., and L.M. wrote the manuscript. All authors discussed the results and commented on the manuscript.

## Competing interests
The authors declare no competing interests.

## Additional information

**Peer review information** *Nature Communications* thanks Otto Muskens, Alessandro Pitanti and the other, anonymous, reviewer(s) for their con-tribution to the peer review of this work. Peer reviewer reports are available.

