## [Peer Review File · Nature Communications]

Gigahertz Optoacoustic Vibration in Sub-5 nm Tip-supported Nano-optomechanical MetasurfaceReviewer #1 (Remarks to the Author):

Dear Editor,

Gao et al. reports on fabrication and experimental characterization of a tip-supported optomechanical metasurface which show light intensity modulation in the GHz range. I think the research results are very interesting, yet the manuscript could be improved before publication. In the following, I am going to discuss few issues which should be addressed in order to have a more clear and impacting paper.

- All the figures need some revisions: it is hard, if not impossible to read the axis labels, legends, etc. The authors should be consistent with the color-code used: for example, "Fit" and "Experiment" have been swapped in panels (b) and (c) of Fig. 2. Also, the "jet" colormap has been used from blue to red in Fig. 1 and 3, while it has been flipped in Fig. 2 (a); this could lead to some confusion. I think the Temperature legend entries of Fig. 5 have been swapped. Finally, in the caption of Fig. 4 there are no (e-i) panels and there is no description of panel (d).

- The experimental results are sound, the only data that puzzles me a little is the spectral map of Fig. 2 (a). I do not understand why the modulation is present only on the red sideband of both the SPP and LSPR resonances, while it looks like it is completely lacking or at least strongly suppressed on the blue sidebands (e.g. around 525nm and 725nm). I would roughly expect the modulated light spectrum to be proportional to the derivative of the reflectivity over the wavelength and from the data of Fig. 1 (e) the resonances seem quite symmetric. Could the authors explain why this is not the case in their experiment ?

- I think it is mandatory to add some discussion of relevant, quantitative metric of the experiment. It seems that the portion of modulated light is less than 1% (from the results of Fig. 2). Could the author be more precise on this and discuss some perspective to increase this number ? Would it be possible to add an estimate of the optomechanical coupling constants ? This would also help to compare this work with similar systems. Also, the authors mention twice a "theoretical limit" for the metasurface modulation capabilities: could they shine some light on that ? It would be interesting to understand what is the main limiting factor in this kind of systems and in their particular realization.

- I think the most important novelty of the report lies in the tip-supported metasurface. Therefore, I believe the readers could appreciate some more details on the fabricated devices. What is the tip size dispersion ? Is there any inhomogeneity in the final etching process ? If that is the case, what is a reasonable area with good homogeneity ? Also, while some results of Fig. 4 are instructive in showing the device properties modification with hole diameters and metasurface thickness, I think it would be extremely important to show at least some simulation of the effect of different tip sizes. How would this affect the optical resonances and the thermal dynamics ?

- Finally, even if the state of the art of optomechanical metasurfaces is quite comprehensive, the authors could also consider citing some relevant work on dielectric optomechanical metasurfaces for polarization modulation, S. Zanotto et al., Adv. Opt. Mat. 8, 1901507 (2020).

Reviewer #2 (Remarks to the Author):

Review report of "Ultrafast Gigahertz Optoacoustic Vibration in Sub-5 nm Tip-Supported Nano-optomechanical Metasurface"

Paper summary:

In this paper, a tip-supported nano-optomechanical metasurface (TSNOM) array is

fabricated using inductively coupled plasma (ICP) etching. With this design, the authors seek to address two main challenges facing nano-optomechanical metasurface structures. The first challenge being the notable energy loss that such structures suffer from, while the second challenge is the nonradiative loss due to structural damage under high pump fluence. The authors have demonstrated the efficacy of this design by using a number of well established experimental and computational methods, including optical reflectance spectroscopy, ultrafast transient reflection (pump-probe) spectroscopy and numerical optical, mechanical and thermal modellings. With this design, the authors were able to experimentally demonstrate high quality factor gigahertz optoacoustic vibrations. It is shown, using mechanical and thermal modelling, that the high-quality factor of the TSNOM array is due to the limited transfer of thermal energy from the metasurface to the underlying sub-5 nm support structure. The authors also demonstrate the structural stability of the TSNOM array by subjecting it to cycles of transient pump-probe spectroscopy, with limited degradation in the signal quality. Another interesting result that was presented in the paper is the linear tunability of the optoacoustic response of the TSNOM array, which is achievable by altering the diameter of the holes on the metastructure.

Given the interesting results, I recommend publishing this work after the author address a number of minor observations:

1 – The values of the colourmap bars in figures 2 (a-c) are illegible. The authors should fix this. In general, the authors should consider increasing the font sizes of all their figures.

2 – Between lines 192 – 193, the authors claim that “the temperature of structure 2 rapidly decays after 10 ps, whereas that of structure 1 remains at a relatively high temperature at 200 ps”. However, according to figure 3 c, it appears that the temperature of structure 1 (tip-supported) is the one that has experienced decay, while structure 2 (substrate-supported) remains relatively unchanged up to 200 ps. The authors should clarify this apparent discrepancy, or otherwise, correct the figure.

3 – While the authors discuss frequency tunability as one of the advantages of TSNOM arrays, it is not clear from this work if the frequency tunability is a unique property of TSNOM arrays, or if similar tunability may also be achieved with the substrate-supported metasurface array. It may be sufficient for the authors to carry out supplemental mechanical simulations for the substrate-supported system to check this.

4 – In the conclusion (line 268), the authors wrote that “Furthermore, the SSNOMS allows precise manipulation of arbitrary structural parameters...”. It is not clear if the authors meant to write TSNOMS here. The authors should check that the correct meaning is conveyed.

5 – The authors should consider removing the “Ultrafast” from the title of the manuscript, as the term “gigahertz” sufficiently conveys the frequency range, and “ultrafast” does not add any additional meaning in the context.

6a- In the abstract, the sentence “The construction of a sub-5 nm tip supporting the semisuspended state intelligently opens the optical input energy channel and closes the mechanical and thermal output energy loss channels” is rather vague at this point. It is not clear what is meant with intelligence here, which is the intelligent part, the design by the team or the device itself? Better remove the term as there is no real intelligence in the system. The closing of mechanical and thermal loss channels is not clear at this point, some effort should be made perhaps with an extra sentence to explain what this means in practice.

6b- Also in the abstract, “The mechanical quality factor Q can be improved by two orders of magnitude compared to the conventional metasurface supported by silicon substrates.” Please specify whether this is an experimental result or following from simulations.

7- Inhomogeneity is an important aspect limiting reproducibility of devices in this frequency range. Could the authors comment on the distribution in both mechanical and optical responses (resonance frequencies, linewidths, spectra) of the devices fabricated. Given the large area of the setup, they are averaging over many unit cells. Could they estimate the impact on e.g. linewidth of the GHz resonances in Figure 4b as well as optical resonances in Fig 1e. Figure 5c suggests very low inhomogeneous broadening but that is for the average frequency, not the local spread. It may be the average is

quite well defined but still there is a natural variation around this average causing the broadening.

8- In particular dephasing of different resonators in the system would cause a more rapid decay.

9- Interband absorption at 311nm seems to be localized spatially quite far away from the tips in the silicon substrate. Explain how this affects the excitation of tip-vibrational modes. Have they considered how changing pumping conditions could modify the efficiency of driving the specific modes. For example would pumping at 725nm be a better choice given the strong near-fields at the tips in Figure 1f? This could perhaps be done with the same laser system but using a different harmonic.

10- Page 5, line 173, specify what is meant by low-frequency modes in this case. Presumably the <1 GHz modes? This is still not very low frequency, just lower than the highest frequency.

Reviewer #3 (Remarks to the Author):

opto acoustic modulation was observed on a nano-membrane suspended on nano-tips. optical properties of metasurface were harnessed to deliver energy by absorption and response of surface was detected in GHz spectral range.

The results are extracted from standard reflectivity measurements and somehow are expected. Deeper physics insights are not presented. it would be useful to show dispersion maps together with experimental results. what different vibration modes exist and how they can be excited. what are energy transfer between modes. can the observation be described by simple oscillator. what we can learn from this metasurface for a particular application.

why this particular geometry of sample was chosen?

Response to Decision Letter

Reviewer #1

General comments:

Gao et al. reports on fabrication and experimental characterization of a tip-supported optomechanical metasurface which show light intensity modulation in the GHz range. I think the research results are very interesting, yet the manuscript could be improved before publication. In the following, I am going to discuss few issues which should be addressed in order to have a more clear and impacting paper.

Reply: We thank the Reviewer for his/her positive comments. In the revised manuscript, we have inserted relevant discussions and added some new experimental as well as simulation data to address these concerns. Detailed responses to all of the points raised by the Reviewer are described below.

Question 1: All the figures need some revisions: it is hard, if not impossible to read the axis labels, legends, etc. The authors should be consistent with the color-code used: for example, "Fit" and "Experiment" have been swapped in panels (b) and (c) of Fig. 2. Also, the "jet" colormap has been used from blue to red in Fig. 1 and 3, while it has been flipped in Fig. 2 (a); this could lead to some confusion. I think the Temperature legend entries of Fig. 5 have been swapped. Finally, in the caption of Fig. 4 there are no (e-i) panels and there is no description of panel (d).

Reply: We thank the reviewer's suggestions. In the revised manuscript, we have corrected the above issues.

1. In the revised manuscript, we have resized the axis labels and legends etc. to make them easier to read.
2. To ensure a consistent color code for the images, we replaced the color of "Fit" and "Experiment" in Fig. 2b of the revised manuscript (page 4, Fig. 2b).
3. According to the Reviewer's suggestion, we have flipped the color legend entry in Fig. 2a-c of the revised manuscript (page 4, Fig. 2a-c).

4. According to the Reviewer's suggestion, we have redrawn the temperature legend entries in Fig. 5b (page 8, Fig. 5b).

5. According to the Reviewer's suggestion, we have added (e) panels in Fig. 4 and added the description of panel (d) (page 7, lines 29-30, and Fig. 4e).

Question 2: The experimental results are sound, the only data that puzzles me a little is the spectral map of Fig. 2 (a). I do not understand why the modulation is present only on the red sideband of both the SPP and LSPR resonances, while it looks like it is completely lacking or at least strongly suppressed on the blue sidebands (e.g. around 525nm and 725nm). I would roughly expect the modulated light spectrum to be proportional to the derivative of the reflectivity over the wavelength and from the data of Fig. 1 (e) the resonances seem quite symmetric. Could the authors explain why this is not the case in their experiment?

Reply: We appreciate the Reviewer's concerns, and we agree that this is a very good point. In the revised manuscript, we have added relevant discussions and explained the reason that modulation is present only in the sideband of the SPP resonance and the LSPR resonance (page 4, lines 41-44; page 5, lines 1; page 5, lines 34-39; and Supporting Information S7). Here, we provide a brief discussion.

As shown in Fig. R1a, the high frequency oscillations are observed in both the red and blue sidebands of the LSPR mode. To explore the reason that modulation is present only in the red and blue sidebands of the LSPR resonance, the reflection spectrum of the structure after in-plane oscillatory deformation was calculated by COMSOL (The structure diagram and COMSOL simulation structure diagram are shown in Fig. R1b, c). Fig. R1d shows the simulation results for the reflection spectra of a TSNOMS before and after the in-plane deformation. As shown in Fig. R1d, the variation in the hole diameter of the TSNOMS has a pronounced modulation effect on the red and blue sidebands of the LSPR resonance, which is in perfect agreement with the LSPR red and blue sideband modulation results shown in Fig. R1a of the previous manuscript. The above simulation results demonstrate that the high-frequency oscillations of the transient signal in the red and blue sidebands of the LSPR mode are caused by

modulation of the spectral dispersion by the periodic in-plane deformation of the nanohole. In response to the Reviewers' comments, we have added a description of the sideband modulation in the revised manuscript and Supporting Information (page 4, lines 41-44; page 5, lines 1; page 5, lines 34-39; and Supporting Information S7).

Fig. R1 (a) Transient reflection spectral map of the TSNOMS for the spectral range between 450 and 830 nm with a delay time up to 7000 ps. (b) The structure diagram of in-plane deformations. (c) COMSOL simulation structure diagram. (d) Simulate the reflectance spectra of in-plane deformations.

As shown in Fig. 2Ra, the relatively low-frequency oscillations are observed in the red sideband of the SPP mode. To explore the reason that modulation is present only in the red sideband of the SPP mode, the reflection spectrum of the structure after out-of-plane oscillatory deformation was calculated by COMSOL (the structure diagram and COMSOL simulation structure diagram are shown in Fig. R2b, c). Fig. R2d shows the simulation results for the reflection spectra of the TSNOMS before and after the out-of-plane deformation. As shown in Fig. R2d, the out-of-plane periodic oscillation mode modulates both the red and blue sidebands of the SPP resonance. In contrast, the modulation is present only on the red sideband of the SPP resonances in Fig. R2a. To investigate the reasons for the conflicting results, we performed the transient dynamic signals of gold nanofilms (Fig. R2e). In Fig. R2e, an obvious transient dynamic signal of the interband transition of gold appears at approximately 510 nm¹. Therefore, we can determine that there is energy competition between the interband transition of gold in the TSNOMS and the SPP resonance. As a result, the transient signal in the blue sideband of the SPP resonance is covered by the transient dynamic signal of the interband transition of gold. Therefore, the transient spectral results show that the modulation is present only on the red sideband of the SPP resonances. In the revised manuscript and Supporting Information, we have added a description of the red

sideband modulation of SPP resonances (page 4, lines 41-44; page 5, lines 1; page 5, lines 34-39; and Supporting Information S7).

Figure R2 (a) Transient reflection spectral map of the TSNOMS for the spectral range between 450 and 830 nm with a delay time up to 7000 ps. (b) The structure diagram of out-of-plane deformations. (c) COMSOL simulation structure diagram. (d) Simulate the reflectance spectra of out-of-plane deformations. (e) Transient reflection spectral map of the gold nanofilm.

Question 3:

I think it is mandatory to add some discussion of relevant, quantitative metric of the experiment. It seems that the portion of modulated light is less than 1% (from the results of Fig. 2). Could the author be more precise on this and discuss some perspective to increase this number? Would it be possible to add an estimate of the optomechanical coupling constants? This would also help to compare this work with similar systems. Also, the authors mention twice a "theoretical limit" for the metasurface modulation capabilities: could they shine some light on that? It would be interesting to understand what is the main limiting factor in this kind of systems and in their particular realization.

Reply: We thank the reviewer’s insightful comments. In the revised manuscript, we have inserted relevant discussions and added some new experimental and simulation data to address these concerns. Here, we provide a brief discussion.

Regarding 1) I think it is mandatory to add some discussion of relevant, quantitative metric of the experiment. It seems that the portion of modulated light is less than 1%

(from the results of Fig. 2). Could the author be more precise on this and discuss some perspective to increase this number? Would it be possible to add an estimate of the optomechanical coupling constants? This would also help to compare this work with similar systems.

The central scientific problem discussed in this manuscript is how to increase the modulation frequency while minimizing the energy loss of an optomechanical system. Therefore, we focus primarily on the modulation frequency and quality factor of an optomechanical metasurface. However, we agree with the Reviewer that a discussion of increasing the portion of modulated light is warranted. Therefore, we have added relevant discussions to the manuscript and Supporting Information (page 5 lines 1-5; Supporting Information S8). Adding an estimate of the optomechanical coupling constants is a constructive suggestion. However, the quantitative assessment of optomechanical coupling constants by transient reflection spectroscopy is still a great challenge. Notably, the portion of modulated light is related to the power density of the pump laser, and thus, we have provided power-dependent transient reflection spectra in the Supporting Information (see Fig. R3 and Supporting Information S8).

Fig. R3 Power-dependent transient reflection spectra.

Regarding 2) Also, the authors mention twice a "theoretical limit" for the metasurface modulation capabilities: could they shine some light on that? It would be interesting to understand what is the main limiting factor in this kind of systems and in their particular

realization.

The “theoretical limit” mentioned in the previous manuscript referred to the fact that the design strategy of a sub-5nm tip-supported metasurface can reduce the energy loss from the substrate to close to the theoretical limit. In response to the Reviewer’s reminder, we have revised the description of the “theoretical limits” and added a more detailed discussion in the revised manuscript (page 7, lines 6-8; page 9, lines 20-22; and Supporting Information S9). Here, we provide a brief discussion.

The conversion of light energy to mechanical energy in an optomechanical system consists of three main energy loss channels: photothermal, structural thermal, and mechanical energy losses. To illustrate the meaning of “the design strategy of a sub-5nm tip-supported metasurface can reduce the energy loss from the substrate to close to the theoretical limit” in the manuscript, we calculated the photothermal conversion, thermal energy and mechanical energy losses for metasurfaces with different contact areas with the substrate using a two-temperature model and the finite element method (see Fig. R4). In terms of photothermal conversion loss, we calculated the variation in electron and lattice temperatures over time at the metasurface under 311-nm laser excitation using the wave optics model of COMSOL and the two-temperature model equation. Fig. R4a shows that the four structures have different quasi-equilibrium temperatures when the electron and phonon temperatures converge due to the different coupling capabilities of the four structures to the 311-nm laser. Fig. R4b shows that as the substrate contact area decreases, the electron-lattice quasi-equilibrium temperature gradually increases, and it reaches a maximum when the nanotips are less than 5 nm wide. This means that the sub-5 nm nanotip array allows the TSNOMS to more easily couple the energy of the pulsed laser into the optomechanical system to excite optoacoustic vibration. In terms of structural thermal energy loss, the thermal distributions in the time domain of the four supported forms of the metasurface were calculated by coupling the solid-state heat transfer model of COMSOL. Fig. R4c, d shows that the sub-5 nm nanotips of Structure 4 minimize the thermal loss from the substrate compared to Structures 1, 2 and 3. For the mechanical energy channel, the geometric strain of the metasurface under the same prestress was simulated by the

COMSOL structural mechanics model for both support methods (see Fig. R3e, f). The sub-5 nm tip-supported metasurface (Structure 4) has the highest strain rate for the same prestress conditions, which indicates that less than the sub-5 nm tip-supported metasurface has the lowest mechanical energy loss for the same pulse stress. Therefore, the numerical theoretical analysis of the three energy loss channels determined that the design strategy of a sub-5nm tip-supported metasurface can reduce the energy loss from the substrate to close to the theoretical limit. The multichannel-loss-mitigating semi-suspended metasurface design strategy can be generalized to performance improvements of most on-chip processed nano-optomechanical systems. And the strategy will drive the development of steady-state metamaterials towards reconfigurability, opening prospects for all-optical operation of nanomechanical systems, reconfigurable nanophotonic devices, optomechanical sensing, and novel nonlinear and self-adaptive photonic functionalities.

Fig. R4 (a) Simulate the electronic and lattice temperature of four supported forms of metasurface by calculating the solid-state heat transfer module with two temperature model (2TM) and the wave optics module of COMSOL. (b) electron-lattice quasi-equilibrium temperature of four supported forms of metasurface. (c) (d) Simulated thermal distribution in the time domain of the four supported forms of the optomechanical metasurface. (e) (f) The simulated geometric strain of the metasurface under the same prestress.

Question 4: I think the most important novelty of the report lies in the tip-supported metasurface. Therefore, I believe the readers could appreciate some more details on the fabricated devices. What is the tip size dispersion? Is there any inhomogeneity in the final etching process? If that is the case, what is a reasonable area with good homogeneity? Also, while some results of Fig. 4 are instructive in showing the device properties modification with hole diameters and metasurface thickness, I think it would be extremely important to show at least some simulation of the effect of different tip sizes. How would this affect the optical resonances and the thermal dynamics?

Reply: We appreciate the Reviewer's comments. In the revised manuscript, we have inserted relevant discussions to address these concerns. Here, we provide a brief discussion.

Regarding 1) I think the most important novelty of the report lies in the tip-supported metasurface. Therefore, I believe the readers could appreciate some more details on the fabricated devices. What is the tip size dispersion? Is there any inhomogeneity in the final etching process? If that is the case, what is a reasonable area with good homogeneity?

We appreciate the Reviewer's comments. In the revised manuscript, we have added a more detailed description of the fabricated devices (page 3, lines 1-7 and Supporting Information S2). The silicon nano-tip arrays of the TSNOMS used in this study were prepared by an inductively coupled plasma (ICP) etching process using gold nanohole arrays as templates. ICP etching allows for the preparation of nanodevices with large areas that are extremely uniform and precise. Thus, the size dispersion of TSNOMS is related only to the homogeneity of gold nanohole array templates. Fig. R5a, b shows SEM images of a large area of the TSNOMS array (relevant images were added to Supporting Information S2). The structure exhibits large areas that are very uniform and aligned, with a relative standard deviation of less than 1% in the hole diameter. As shown in Fig. R5c-f, the silicon nanotip arrays also exhibit great homogeneity over a large area due to the extremely homogeneous nanohole array templates.

Fig. R5 SEM image of the TSNOMS.

Regarding 2) Also, while some results of Fig. 4 are instructive in showing the device properties modification with hole diameters and metasurface thickness, I think it would be extremely important to show at least some simulation of the effect of different tip sizes. How would this affect the optical resonances and the thermal dynamics?

We appreciate the Reviewer's suggestion, and we agree that this is a very good point. According to the Reviewer's reminder, we have performed numerical theoretical calculations of the optical resonances, thermal dynamics, and steady-state mechanical properties of metasurfaces with different nanotip sizes and have added the relevant content to the Supporting Information S9.

Question 5: Finally, even if the state of the art of optomechanical metasurfaces is quite comprehensive, the authors could also consider citing some relevant work on dielectric optomechanical metasurfaces for polarization modulation, S. Zanotto et al., *Adv. Opt. Mat.* **8**, 1901507 (2020).

Reply: Many thanks for the valuable suggestion. According to the Reviewer's reminder, in the revised manuscript, we have added references to relevant articles (page 10, highlighted in yellow).

Reference

1. Cooper B.R., Ehrenreich H., Philipp H.R. Optical Properties of Noble Metals. II. *Phys. Rev.* **138**, A494 (1965).

Reviewer #2

General comments:

In this paper, a tip-supported nano-optomechanical metasurface (TSNOM) array is fabricated using inductively coupled plasma (ICP) etching. With this design, the authors seek to address two main challenges facing nano-optomechanical metasurface structures. The first challenge being the notable energy loss that such structures suffer from, while the second challenge is the nonradiative loss due to structural damage under high pump fluence. The authors have demonstrated the efficacy of this design by using a number of well established experimental and computational methods, including optical reflectance spectroscopy, ultrafast transient reflection (pump-probe) spectroscopy and numerical optical, mechanical and thermal modellings. With this design, the authors were able to experimentally demonstrate high quality factor gigahertz optoacoustic vibrations. It is shown, using mechanical and thermal modelling, that the high-quality factor of the TSNOM array is due to the limited transfer of thermal energy from the metasurface to the underlying sub-5 nm support structure. The authors also demonstrate the structural stability of the TSNOM array by subjecting it to cycles of transient pump-probe spectroscopy, with limited degradation in the signal quality. Another interesting result that was presented in the paper is the linear tunability of the optoacoustic response of the TSNOM array, which is achievable by altering the diameter of the holes on the metastructure.

Given the interesting results, I recommend publishing this work after the author address a number of minor observations:

Reply: We thank the Reviewer for his/her positive comments. In the revised manuscript, we have inserted relevant discussions and added some new experimental as well as simulation data to address these concerns. Detailed responses to all of the points raised by the Reviewer are described below.

Question 1: The values of the colourmap bars in figures 2 (a-c) are illegible. The authors should fix this. In general, the authors should consider increasing the font sizes

of all their figures.

Reply: We appreciate the Reviewer's suggestion. We have modified the colormap bars in Fig. 2a-c and increased the font size in all figures (page 3, Fig 1; page 4, Fig 2; page 6, Fig 3; page 7, Fig 4; page 8, Fig 5).

Question 2: Between lines 192-193, the authors claim that “the temperature of structure 2 rapidly decays after 10 ps, whereas that of structure 1 remains at a relatively high temperature at 200 ps”. However, according to figure 3 c, it appears that the temperature of structure 1 (tip-supported) is the one that has experienced decay, while structure 2 (substrate-supported) remains relatively unchanged up to 200 ps. The authors should clarify this apparent discrepancy, or otherwise, correct the figure.

Reply: We thank the reviewers for their attention to detail in noting this error. We incorrectly marked the legend in the previous manuscript, and we have corrected in the revised manuscript (page 6, Fig 3b).

Question 3: While the authors discuss frequency tunability as one of the advantages of TSNOM arrays, it is not clear from this work if the frequency tunability is a unique property of TSNOM arrays, or if similar tunability may also be achieved with the substrate-supported metasurface array. It may be sufficient for the authors to carry out supplemental mechanical simulations for the substrate-supported system to check this.

Reply: We thank the Reviewer for this suggestion. In general, the eigenfrequencies of mechanical oscillations of nanostructures are linearly related to geometrical parameters. Theoretically, both nanotip-supported and substrate-supported metasurfaces can be modulated for frequency by changing the geometrical parameters. However, as shown in Fig. R1, the transient dynamics signal of mechanical oscillations is completely undetectable on the substrate-supported metasurface due to the enormous losses from the substrate. The frequency tunability of mechanical oscillations presupposes that the transient dynamics signal can be observed. In this sense, frequency tunability is a unique property of TSNOM arrays. According to the Reviewer's reminder, we have added relevant discussions to clarify this point (please see page 7, lines 41-44).

Fig. R1 Transient reflection spectra of the optomechanical metasurface supported on a monocrystalline silicon substrate and the silicon nanotip arrays.

Question 4: In the conclusion (line 268), the authors wrote that “Furthermore, the SSNOMS allows precise manipulation of arbitrary structural parameters...”. It is not clear if the authors meant to write TSNOMS here. The authors should check that the correct meaning is conveyed.

Reply: We thank the reviewers for their attention to detail in noting this error, we have changed SSNOMS to TSNOMS in the revised manuscript (page 9, line 14).

Question 5: The authors should consider removing the “Ultrafast” from the title of the manuscript, as the term “gigahertz” sufficiently conveys the frequency range, and “ultrafast” does not add any additional meaning in the context.

Reply: Thanks for the suggestion. For clarity, we have removed “Ultrafast” from the title of the manuscript (page 1, line 1).

Question 6a: In the abstract, the sentence “The construction of a sub-5 nm tip supporting the semisuspended state intelligently opens the optical input energy channel and closes the mechanical and thermal output energy loss channels” is rather vague at this point. It is not clear what is meant with intelligence here, which is the intelligent part, the design by the team or the device itself? Better remove the term as there is no real intelligence in the system. The closing of mechanical and thermal loss channels is not clear at this point, some effort should be made perhaps with an extra sentence to

explain what this means in practice.

Reply: We appreciate the Reviewer's comment. For clarity, we have removed "intelligently" and added the following description on page 1, lines 19-24:

"we demonstrated a high-quality 5 GHz optoacoustic vibration and ultrafast optomechanical all-optical manipulation in a sub-5 nm tip-supported nano-optomechanical metasurface (TSNOMS). The physical rationale is that the design of the semi-suspended metasurface supported by nanotips of less than 5 nm enhances the optical energy input into the metasurface and closes the mechanical and thermal output loss channels, result in dramatically improvement of the optomechanical conversion efficiency and oscillation quality of the metasurface"

Question 6b: Also in the abstract, "The mechanical quality factor Q can be improved by two orders of magnitude compared to the conventional metasurface supported by silicon substrates." Please specify whether this is an experimental result or following from simulations.

Reply: We appreciate the Reviewer's comment. Based on the experimental results in Fig. 3a in the manuscript, we can calculate that the mechanical quality factor Q of the silicon substrate-supported metasurface is 0, while the mechanical quality factor Q of the TSNOMS optical machine oscillation is 37. Therefore, we believe that the expression "two orders of magnitude" is indeed inaccurate, and we have revised the expression in the abstract (page 1, lines 19-24).

Question 7: Inhomogeneity is an important aspect limiting reproducibility of devices in this frequency range. Could the authors comment on the distribution in both mechanical and optical responses (resonance frequencies, linewidths, spectra) of the devices fabricated. Given the large area of the setup, they are averaging over many unit cells. Could they estimate the impact on e.g. linewidth of the GHz resonances in Figure 4b as well as optical resonances in Fig 1e. Figure 5c suggests very low inhomogeneous broadening but that is for the average frequency, not the local spread. It may be the average is quite well defined but still there is a natural variation around this average

causing the broadening.

Reply: The Reviewer’s note on “average frequency” is a valuable reminder, and we are grateful for the Reviewer’s advice. In this study, a pulsed laser with Gaussian diameter of 1 mm was used to excite the vibration modes of the samples, so the transient spectrum is the average data of many units. This inevitably leads to natural variation around this average, which causes broadening. Fig. 1e and Fig. 4b show that the full width at half maximum (FWHM) of the experimental spectrums are larger than that of the simulated results. This part of the broadening is mainly due to inhomogeneous broadening. In addition to the inhomogeneous broadening effect, the radiation loss, non-radiation loss, out-of-phase oscillation, and jitter of the optical path of the experimental equipment affect the FWHM of the mechanical and optical spectrums. Thus, it is very difficult to quantify the effect of inhomogeneous broadening on FWHM. To minimize the inhomogeneous broadening effect of the sample, improving the precision and uniformity of the prepared sample is the best option. Fig. R2 shows that our samples are extremely uniform over a large area. However, the inhomogeneous broadening effect is still inevitable. Following the reminder from the Reviewer, we have added a more detailed description of the inhomogeneous broadening effect to the revised manuscript (page 3, lines 1-7 and page 5, lines 27-30).

Fig. R2 SEM image of the TSNOMS.

Question 8: In particular dephasing of different resonators in the system would cause a more rapid decay.

Reply: We thank the reviewer’s insightful comments. The particular dephasing of different resonators affects the oscillation decay time and the quality factor of the optomechanical metasurface. Based on the Reviewer’s reminder, we have added a discussion of the issue of particular dephasing of different resonators to the revised

manuscript (page 5, lines 27-30).

Question 9: Interband absorption at 311nm seems to be localized spatially quite far away from the tips in the silicon substrate. Explain how this affects the excitation of tip-vibrational modes. Have they considered how changing pumping conditions could modify the efficiency of driving the specific modes. For example would pumping at 725nm be a better choice given the strong near-fields at the tips in Figure 1f? This could perhaps be done with the same laser system but using a different harmonic.

Regarding 1) Interband absorption at 311nm seems to be localized spatially quite far away from the tips in the silicon substrate. Explain how this affects the excitation of tip-vibrational modes.

Reply: We thank the Reviewer for this comment. In this manuscript, the 311 nm pump laser drives the in-plane and out-of-plane vibrational modes of the gold nanoholes of the TSNOMS rather than the tip-vibrational modes. (Fig. R3a shows the TSNOMS eigenmodes calculated by COMSOL.) Therefore, we need to localize the optical field to the area near the gold nanohole arrays rather than the silicon nanotips. In the previous manuscript, we explained the basic principle of the 311 nm pump laser driving the optoacoustic vibration of the TSNOMS (please see page 3, lines 41-44; page 4, lines 1-5 and Supporting Information S4). To avoid misunderstanding by readers, we have provided a more detailed discussion of the mechanism of optoacoustic vibration excited by interband transitions to the revised manuscript (see page4, lines 37-38; Supporting Information S6 and Supporting Information S4). Here, we provide a brief discussion.

When a beam of light is reflected by the substrate, the reflected light is superimposed on the incident light to form a standing wave.

Incident wave:

$$E_{in} = E_0 e^{i(kz - \omega t)}$$

Reflected wave:

$$E_r = r E_0 e^{i(-kz - \omega t)}$$

The form of the field of the superimposed incident and reflected light is as follows:

$$E = E_0 e^{i(kz - \omega t)} + r E_0 e^{i(-kz - \omega t)}$$

$$E=E_0((1+r_1)\cos(kz)+ir_2\cos(kz)+i(1-r_1)\sin(kz)+r_2\sin(kz))(\cos(\omega t)-isin(\omega t))$$

After retaining the real part:

$$E=E_0((1+r_1)\cos(kz)+r_2\sin(kz))\cos(\omega t)+(r_2\cos(kz)+(1-r_1)\sin(kz))\sin(\omega t)$$

$$E=E_0\sqrt{(1+|r|^2+2|r|\cos(2kz+\varphi_1))}\sin(\omega t+\varphi)$$

The above equation shows that the electric field intensity is a function of spatial position z , and the distance between the antinode and the node of a standing wave is $\Delta z = \lambda/4$. According to the above derivation combined with the reflection spectrum shown in Fig. R3b, the distribution of the metasurface electric field was calculated by COMSOL when the incident wavelength was 215, 311, 257, and 434 nm (Fig. R3c). As shown in Fig. R3c, the distance between the electric field ventral and the electric field nodes is consistent with the above corollary. Due to the support of the silicon nanotips, when the incident light wavelength is 215 or 311 nm, the diffraction wave junction covers the plane where the gold metasurface is located. When the incident wavelength is 257 or 434 nm, the distance between the electric field node and the gold metasurface is exactly one-eighth of the wavelength. Therefore, when the energy of the incident laser is higher than the interband transition energy of gold, the more overlap between the electric field node and the gold metasurface, and the higher absorption rate of the metasurface. Based on the above discussion, the absorption dip at a wavelength of 311 nm is confirmed to originate from the interband transition absorption of gold.

Fig. R3 (a) Simulated mechanical eigenmodes at selected frequencies. (b) The measured reflection spectra of the TSNOMS (black lines) and the simulated transmission spectra calculated by COMMSOL (red dotted line). (c) The simulated electric field intensity distribution of the TSNOMS.

In this approach, the 311 nm pump pulse was used to selectively excite the conduction electrons of the metal by interband transition absorption, followed by fast Auger relaxation of the photoexcited electrons¹ (see step 1 in Fig. R4). Except for very

short time delays, the excitation processes lead to similar nonequilibrium conduction electron distribution that thermalizes by electron-electron scattering² (see step 2 in Fig. R4). The energy given to the electrons is subsequently damped to the lattice by electron-vibration interactions (see step 3 in Fig. R4). The time evolutions of the electronic, T_e , and lattice, T_l , temperatures are described by the rate equation system (two-temperature model)³⁻⁵:

$$C_e(T_e) \frac{dT_e}{dt} = \nabla \cdot (\kappa_e \nabla T_e) - G(T_e) \times (T_e - T_l) + S(t) \quad (1)$$

$$C_l(T_l) \frac{dT_l}{dt} = (\kappa_p \nabla T_l) \nabla \cdot G(T_e) \times (T_e - T_l) \quad (2)$$

Excitation of the metal electrons and electron-lattice thermalization leads to hot nano-objects that cool down to initial temperature by energy transfer to their surroundings (matrix or substrate) and heat diffusion in the latter⁶. This takes place on a timescale of typically a few tens to hundred picoseconds depending on the environment, the size of the object, and their coupling. Direct electron heating by the pump pulse and fast indirect heating of the lattice by electron-lattice energy transfer both impose dilation of the nano-object and launch its acoustic vibrations⁷ (see step 4 in Fig. R4). Heating of the electrons and lattice of a nano-object and launching of its acoustic vibration modify its dielectric function. This translates into a time-dependent modification of its optical response that can be followed using femtosecond transient spectroscopy (see Supporting Information S6 for details of the time domain physical process).

Fig. R4 Main time-domain processes after selective electron heating at temperature of the electron gas of the TSNOMS by a femtosecond pulse: electron-lattice thermalization, acoustic vibrations, and thermal and acoustic energy damping to the environment.

Regarding 2) Have they considered how changing pumping conditions could modify

the efficiency of driving the specific modes. For example would pumping at 725nm be a better choice given the strong near-fields at the tips in Figure 1f? This could perhaps be done with the same laser system but using a different harmonic.

Reply: Many thanks to the Reviewer for this reminder. In fact, the strong near-fields at 725 nm in Fig. 1f is localized near the gold nanoholes rather than the silicon nanotips. To avoid misunderstanding, we have added the 3D electric field distribution at 725 nm. Fig. R5b shows that the resonance absorption at 725 nm results from the dipolar surface plasmon resonance (SPR) of the nanoholes and that there is no significant strong near-field near the silicon nanotips.

Fig. R5 (a) The simulated electric field intensity distribution of the TSNOMS. **(b)** The 3D electric field intensity distribution at 725 nm

It is a very good point to change the pump conditions to modify the efficiency of driving the specific modes, but it was not clearly observed in our experimental results. Based on the Reviewer's reminder, we modulated the pump laser to 725 nm by optical parametric amplification (OPA) and performed transient spectroscopy on the TSNOMS (see Fig. R6). Fig. R6a shows that the 725nm pump laser heavily covers the transient spectral signal at approximately 725 nm. Although an oscillation signal was observed at 765 nm (see Fig. R6b), the quality of the oscillation was significantly worse compared to that of the 311nm pump laser. Therefore, 311 nm is a better choice than 725 nm as the source of a pump laser.

Fig. R6 (a) Transient reflection spectral map of the TSNOMS for the pump at 725 nm. (b) Transient reflection spectra of the optomechanical metasurface for probe at 765 nm

Question 10: Page 5, line 173, specify what is meant by low-frequency modes in this case. Presumably the <1 GHz modes? This is still not very low frequency, just lower than the highest frequency.

Reply: Fully agree with the Reviewer. In the previous manuscript, we incorrectly defined 0.3 GHz modes as low-frequency modes. To avoid ambiguity, we have changed the low-frequency modes in the manuscript to out-of-plane oscillation modes (page 5, lines 33-38).

Reference

1. Knoesel E. et al. Ultrafast dynamics of hot electrons and holes in copper: Excitation, energy relaxation, and transport effects. *Phys. Rev. B* **57**, 12812-12824 (1998).
2. Voisin C. et al. Size-dependent electron–electron interactions in metal nanoparticles. *Phys. Rev. Lett.* **85**, 2200-2203 (2000).
3. Block A. et al. Tracking ultrafast hot-electron diffusion in space and time by ultrafast thermomodulation microscopy. *Sci. Adv.* **5**, 8965 (2019).
4. Brown A. M. et al. Ab initiophonon coupling and optical response of hot electrons in plasmonic metals. *Phys. Rev. B* **94**, 075120 (2016).
5. Schirato A. et al. Transient optical symmetry breaking for ultrafast broadband dichroism in plasmonic metasurfaces. *Nat. Photon.* **14**, 723-727 (2020).
6. Juvé V. et al. Cooling dynamics and thermal interface resistance of glass-embedded metal nanoparticles. *Phys. Rev. B* **80**, 195406 (2009).

7. Aurelien C. et al. Acoustic vibrations of metal nano-objects: Time-domain investigations. *Phys. Rep.* **549**, 1-43 (2015).

Reviewer #3

General comments:

opto acoustic modulation was observed on a nano-membrane suspended on nano-tips. optical properties of metasurface were harnessed to deliver energy by absorption and response of surface was detected in GHz spectral range.

The results are extracted from standard reflectivity measurements and somehow are expected. Deeper physics insights are not presented. it would be useful to show dispersion maps together with experimental results. what different vibration modes exist and how they can be excited. what are energy transfer between modes. can the observation be described by simple oscillator. what we can learn from this metasurface for a particular application.

why this particular geometry of sample was chosen?

Reply: We thank the Reviewer for the comments. In the following responses, we have implemented more details and descriptions of the new insights and broad appeal of the manuscript to address the Reviewer's concerns. Detailed responses to all of their points of his (or hers) are described below.

Regarding 1) The results are extracted from standard reflectivity measurements and somehow are expected. Deeper physics insights are not presented.

We thank the Reviewer for the comments. To provide more clarity on the physical insights in the manuscript, we have added more detailed discussion in the revised manuscript and rewritten the abstract to highlight the physical insights (please see page 1, lines 19-24; page 4, lines 37-38; page 7, lines 6-8; Supporting Information S6; Supporting Information S9; Regarding 2 and Regarding 3). Here, we provide a brief discussion.

For nanomechanical systems, it has been an insurmountable challenge to provide

all-optical manipulation of nanomechanical systems and reconfigurable nanophotonic devices with ultralow loss and ultrahigh modulation depth. In this manuscript, we have overcome these challenges for the first time by proposing a new physical model of multichannel loss mitigation. The design of the semi-suspended metasurface supported by nanotips of less than 5 nm cleverly enhances the optical energy input into the metasurface and closes the mechanical and thermal output loss channels of the metasurface, resulting in orders of magnitude increase in the optomechanical modulation capability and mechanical quality factor of the metasurface. The multichannel-loss-mitigating semi-suspended metasurface design strategy can be generalized to performance improvements of most on-chip processed nano-optomechanical systems. The strategy will drive the development of steady-state metamaterials towards transient reconfigurability, opening prospects for all-optical operation of nanomechanical systems, reconfigurable nanophotonic devices, optomechanical sensing, and novel nonlinear and self-adaptive photonic functionalities.

Regarding 2) what different vibration modes exist and how they can be excited. what are energy transfer between modes. can the observation be described by simple oscillator.

We thank the Reviewer for these comments. In the revised manuscript, we have added a more detailed discussion of the vibration modes and how they can be excited (please see page 5, lines 34-39; Supporting Information S6). In addition, we have drawn physical images of the energy relaxation process that excites optoacoustic vibrations to illustrate the ultrafast energy transfer processes in more detail (please see page 4, lines 37-38; Supporting Information S6). In the manuscript, the theoretical calculation results demonstrate that the periodic modulation of the transient spectrum originates from the periodic deformation of the structures. Thus, the periodic modulation of transient spectral signals can be quantitatively described by the damped harmonic oscillator (DHO) model. Detailed responses to all of the points are described below.

In this manuscript, there are three main optical resonance modes in TSNOMS, namely, the interband transition absorption mode at 311 nm, the SPP mode at 532 nm,

and the dipolar surface plasmon resonance (SPR) of the nanoholes at 725 nm. In this approach, the 311 nm pump pulse was used to selectively excite the conduction electrons of the metal by interband transition absorption, followed by fast Auger relaxation of the photoexcited electrons¹ (see step 1 in Fig. R1). Except for very short

Fig. R1 Main time-domain processes after selective electron heating at temperature of the electron gas of the TSNOMS by a femtosecond pulse: electron-lattice thermalization, acoustic vibrations, and thermal and acoustic energy damping to the environment.

time delays, the excitation processes lead to similar distributions of nonequilibrium conduction electrons that thermalizes by electron-electron scattering² (see step 2 in Fig. R1). The energy given to the electrons is subsequently damped by the lattice by means of electron-vibration interactions (see step 3 in Fig. R1). The time evolutions of the electronic (T_e) and lattice (T_l) temperatures are described by the rate equation system (two-temperature model)³⁻⁵:

$$C_e(T_e) \frac{dT_e}{dt} = \nabla \cdot (\kappa_e \nabla T_e) - G(T_e) \times (T_e - T_l) + S_e(t) \quad (1)$$

$$C_l(T_l) \frac{dT_l}{dt} = (\kappa_p \nabla T_l) \nabla \cdot - G(T_e) \times (T_e - T_l) \quad (2)$$

Excitation of the metal electrons and electron-lattice thermalization leads to hot nano-objects that cool to the initial temperature by energy transfer to their surroundings (matrix or substrate) and heat diffusion in the latter⁶. This takes place on a timescale from typically a few tens to hundred picoseconds depending on the environment, the size of the object, and their coupling. Direct electron heating by the pump pulse and fast indirect heating of the lattice by electron-lattice energy transfer both dilate the nano-object and initiate its acoustic vibrations⁷ (see step 4 in Fig. R1). Heating the electrons and lattice of a nano-object and initiating its acoustic vibration modify its

dielectric function. This translates into a time-dependent modification of its optical response that can be followed using femtosecond transient spectroscopy (see Fig. R2a). According to the experimental data and theoretical model analysis in Fig. R2f-e, two intrinsic vibration modes were excited, namely, the out-of-plane mechanical vibration mode and the in-plane mechanical vibration mode respectively (For a more detailed discussion please see page 5, lines 34-39; page 4, lines 37-38; Supporting Information S6).

Fig. R2 Transient reflection spectral map of the TSNOMS for the spectral range between 450 and 830 nm with a delay time up to 7000 ps. **b**, Transient reflection spectral map for the spectral range between 785 and 830 nm and $\Delta R/R$ kinetics at 798 nm. **c**, Transient reflection spectral map for the spectral range between 630 and 668 nm and $\Delta R/R$ kinetics at 629 nm. **d**, Fast Fourier transform (FFT) spectra of the $\Delta R/R$ kinetics at 629 and 798 nm. **e**, Simulated frequency spectra of displacements sampled at a point (indicated by a red dot) on the edge of a nanohole. **f**, Simulated mechanical eigenmodes at selected frequencies.

Regarding 3) what we can learn from this metasurface for a particular application. why this particular geometry of sample was chosen?

We thank the Reviewer for these comments. To clarify the tremendous value of TSNOMS in particular applications and the significance of choosing this particular geometry, in the revised manuscript, we have added more detailed discussion to clarify these points (please see page 1, lines 28-33; page 7, lines 6-8 and Supporting Information S9). Here, we provide a brief discussion.

In the nano-optomechanical system, the conversion of light energy to mechanical

energy in an optomechanical system consists of three main energy loss channels: photothermal, structural thermal, and mechanical energy losses. To illustrate the reasons for designing this particular geometry of sample, we calculated the photothermal conversion, thermal energy and mechanical energy losses for metasurfaces with different contact areas with the substrate using a two-temperature model and the finite element method (Fig. R2). In terms of photothermal conversion loss, we calculated the variation in electron and lattice temperatures over time at the metasurface under 311-nm laser excitation using the wave optics model of COMSOL and the two-temperature model equation. Fig. R3a shows that the four structures have different quasi-equilibrium temperatures when the electron and phonon temperatures converge due to the different coupling capabilities of the four structures to the 311-nm laser. Fig. R3b shows that as the substrate contact area decreases, the electron-lattice quasi-equilibrium temperature gradually increases, and it reaches a maximum when the nanotips are less than 5 nm wide. This means that the sub-5 nm nanotip array allows the TSNOMS to more easily couple the energy of the pulsed laser into the optomechanical system to excite optoacoustic vibration. In terms of structural thermal energy loss, the thermal distributions in the time domain of the four supported forms of the metasurface were calculated by coupling the solid-state heat transfer model of COMSOL. Fig. R3c, d shows that the sub-5 nm nanotips of Structure 4 minimize the thermal loss from the substrate compared to Structures 1, 2 and 3. For the mechanical energy channel, the geometric strain of the metasurface under the same prestress was simulated by the COMSOL structural mechanics model for both support methods (see Fig. R3e, f). The sub-5 nm tip-supported metasurface (Structure 4) has the highest strain rate for the same prestress conditions, which indicates that less than the sub-5 nm tip-supported metasurface has the lowest mechanical energy loss for the same pulse stress. Therefore, the numerical theoretical analysis of the three energy loss channels determined that the sub-5 nm nanotip array can reduce the energy losses from the substrate to be infinitely close to the theoretical limit.

The multichannel-loss-mitigating semi-suspended metasurface design strategy can be generalized to performance improvements of most on-chip processed nano-

optomechanical systems. And the strategy will drive the development of steady-state metamaterials towards reconfigurability, opening prospects for all-optical operation of nanomechanical systems, reconfigurable nanophotonic devices, optomechanical sensing, and novel nonlinear and self-adaptive photonic functionalities.

Fig. R3 (a) Simulate the electronic and lattice temperature of four supported forms of metasurface by calculating the solid-state heat transfer module with two model (2TM) and the wave optics module of COMSOL. (b) electron-lattice quasi-equilibrium temperature of four supported forms of metasurface. (c) (d) Simulated thermal distribution in the time domain of the four supported forms of the optomechanical metasurface. (e) (f) The simulated geometric strain of the metasurface under the same prestress.

Reference

1. Knoesel E. et al. Ultrafast dynamics of hot electrons and holes in copper: Excitation, energy relaxation, and transport effects. *Phys. Rev. B* **57**, 12812-12824 (1998).
2. Voisin C. et al. Size-dependent electron–electron interactions in metal nanoparticles. *Phys. Rev. Lett.* **85**, 2200-2203 (2000).
3. Block A. et al. Tracking ultrafast hot-electron diffusion in space and time by ultrafast thermomodulation microscopy. *Sci. Adv.* **5**, 8965 (2019).
4. Brown A. M. et al. Ab initiophonon coupling and optical response of hot electrons in plasmonic metals. *Phys. Rev. B* **94**, 075120 (2016).
5. Schirato A. et al. Transient optical symmetry breaking for ultrafast broadband

dichroism in plasmonic metasurfaces. *Nat. Photon.* **14**, 723-727 (2020).

6. Juvé V. et al. Cooling dynamics and thermal interface resistance of glass-embedded metal nanoparticles. *Phys. Rev. B* **80**, 195406 (2009).

7. Aurelien C. et al. Acoustic vibrations of metal nano-objects: Time-domain investigations. *Phys. Rep.* **549**, 1-43 (2015).

Reviewer #1 (Remarks to the Author):

Dear Editor,

I have carefully read the revised version of the manuscript from Gao et al. and I can see that the authors have addressed all the issues I pointed out in the original manuscript. I am glad to recommend the publication of the manuscript in Nature Communications

Reviewer #2 (Remarks to the Author):

The authors have presented a detailed response to all reviewer comments which in my opinion addresses all points raised. In particular it is appreciated that they have done some extra measurements to investigate the response when pumping at 725nm. I think altogether it is a high quality paper with interesting results and recommend publication in its current form. With my apologies for delay in reviewing this paper.

Reviewer #3 (Remarks to the Author):

review remarks have been answered
what is the amplitude of the surface vibration in theory for the ideal structure and in experiment?
how it compares with other metasurfaces
<https://pubs.acs.org/doi/abs/10.1021/acs.nanolett.7b02663>

Response to referees

Reviewer #1

I have carefully read the revised version of the manuscript from Gao et al. and I can see that the authors have addressed all the issues I pointed out in the original manuscript. I am glad to recommend the publication of the manuscript in Nature Communications.

Reply: Thanks for the reviewer's very positive and encouraging comments.

Reviewer #2

The authors have presented a detailed response to all reviewer comments which in my opinion addresses all points raised. In particular it is appreciated that they have done some extra measurements to investigate the response when pumping at 725nm. I think altogether it is a high quality paper with interesting results and recommend publication in its current form. With my apologies for delay in reviewing this paper.

Reply: Thanks for the reviewer's very positive and encouraging comments.

Reviewer #3

review remarks have been answered. What is the amplitude of the surface vibration in theory for the ideal structure and in experiment? How it compares with other metasurfaces <https://pubs.acs.org/doi/abs/10.1021/acs.nanolett.7b02663>

Reply: Thanks for the reviewer's comments. In principle, the amplitude of the surface vibration is related to the geometrical parameters and material properties of the surface, the power of the laser, the spot size of the pump laser, the ambient temperature, the pressure, and other factors. The amplitude of surface vibrations cannot generally be directly quantified by transient spectroscopy experiments. In experiments, the amplitude of the surface vibration is linearly related to the modulation depth of the transient reflection spectroscopy. As shown in Figure 2 of the manuscript, The $\Delta R/R$ value of up to 0.55% indirectly reflects the maximum amplitude of the surface vibration at a pump fluence of 193.8 $\mu\text{J}/\text{cm}^2$. We have made necessary changes to the expressions in Supporting Information S7 (See supporting information S7, lines 1, 2).

Compared with the paper (Reference 28 in the manuscript) mentioned by the reviewers, the quality factor and vibration time of our metasurface are improved by more than a factor of 4 and the modulation depth is improved by nearly 2 orders of magnitude.